# Reproductive Hormones Mediate Intestinal Microbiota Shifts during Estrus Synchronization in Grazing Simmental Cows

**DOI:** 10.3390/ani12141751

**Published:** 2022-07-07

**Authors:** Donglin Wu, Chunjie Wang, Huasai Simujide, Bo Liu, Zhimeng Chen, Pengfei Zhao, Mingke Huangfu, Jiale Liu, Xin Gao, Yi Wu, Xiaorui Li, Hao Chen, Aorigele Chen

**Affiliations:** 1College of Animal Science, Inner Mongolia Agricultural University, Hohhot 010018, China; 18047182047@163.com (D.W.); smjd_2010@163.com (H.S.); lb15754882650@163.com (B.L.); c1452630556@163.com (Z.C.); zhaopengfei2333@163.com (P.Z.); hfhuahua2022@163.com (M.H.); nmnddky0531@126.com (Y.W.); lxr15048186287@163.com (X.L.); chenhao9781@126.com (H.C.); 2College of Veterinary Medicine, Inner Mongolia Agricultural University, Hohhot 010018, China; ljl1727757400@163.com (J.L.); gaoxin102104@163.com (X.G.)

**Keywords:** intestinal microbiota, estrus synchronization, reproductive hormones, estradiol, 16S rRNA sequencing, grazing Simmental cows

## Abstract

**Simple Summary:**

Characterization of the microbiota in livestock animals is of great interest for improving reproductive performances. Shifts in vaginal and uterine microbiota are well-studied, but the intestinal microbiota in ruminants during estrus is largely unknown. The intestinal microbiota of grazing Simmental cows undergoing estrus synchronization was studied in this work. The structure, composition, and function of the intestinal microbiota shifted during this process, and these shifts were mediated by reproductive hormones.

**Abstract:**

To study shifts in the intestinal microbiota during estrus synchronization in ruminants, we characterized the intestinal microbiota in grazing Simmental cows and the possible mechanism that mediates this shift. Fourteen postpartum Simmental beef cows were synchronized beginning on day 0 (D0) with a controlled internal release device (CIDR), and cloprostenol was injected on D9 when the CIDR was withdrawn. Synchronization ended with timed artificial insemination on D12. Serum and rectal samples harvested on D0, D9, and D12 were analyzed to assess the reproductive hormones and microbiota. Reproductive hormones in the serum of the host were measured using enzyme-linked immunosorbent assay. The microbiota was characterized using 16S rRNA sequencing of the V3–V4 hypervariable region, alpha diversity and beta diversity analyses (principal coordinate analysis, PCoA), cladogram of the linear discriminant analysis effect size (LEfSe) analysis, and microbiota function analysis. Levels of the reproductive hormones, except gonadotropin-releasing hormone (*p* > 0.05), shifted among D0, D9, and D12 (*p* < 0.05). Decreased community diversity (Chao1 and ACE) was observed on D12 compared with D0 (*p* < 0.05). The beta diversity (PCoA) of the microbiota shifted markedly among D0, D9, and D12 (*p* < 0.05). The LEfSe analysis revealed shifts in the intestinal microbiota communities among D0, D9, and D12 (*p* < 0.05 and LDA cutoff >3.0). The KEGG pathway analysis showed that carbohydrate metabolism, genetic information and processing, the excretory system, cellular processes and signaling, immune system diseases, and the metabolism were altered (*p* < 0.05). Reproductive hormones (especially estradiol) were correlated with the alpha diversity indices, beta diversity indices, and an abundance of biomarkers of the shifting intestinal microbiota (*p* < 0.05). In conclusion, the structure, composition, and function of the intestinal microbiota were shifted during estrus synchronization in a grazing Simmental cow model, and these shifts were mediated by reproductive hormones.

## 1. Introduction

Reproduction affects the profitability and turnover rate of farms, particularly cattle farms, which are negatively impacted by the low fertility and high abortion rates of cattle [1]. Reproductive technologies, genetic progress, and enhancement of the management of cattle have been shown to positively improve the reproductive performance [2]. The use of protocols to synchronize estrus and ovulation, leading to timed artificial insemination, has contributed to the reproductive management of cattle.

The microbiota, especially in the intestine, coexists with the host for mutual beneficial purposes and is now considered a virtual organ with properties that are integral to the host’s endocrine, metabolic, and immune systems [3]. Synchronized estrus protocols have been shown to change the vaginal microbiota [4] and cause shifts in the uterine microbiota [5]. Studies on broilers [6], mice [7], and sows [8] have confirmed that the intestinal microbiota positively affect the reproductive performance. These studies underline the importance of the effects of the intestinal microbiota on the host and highlight that shifts in the intestinal microbiota occur related to changes in the female reproductive state. However, intestinal microbiota-host interactions during estrus synchronization are largely unknown in ruminants, and the less explored intestinal microbiota of female cattle may also provide insights that help explain the reproductive success and failure during synchronization.

Bidirectional interactions between the microbiome and the hormonal system have also been observed, particularly involving the sex hormone estrogen [9,10,11]. In fact, sex hormone manipulation during periods of early development has been shown to alter the intestinal microbiota in studies in humans and mice [9,12]. Sex hormones directly modulate the microbial metabolism through steroid receptors, including estrogen receptor beta, particularly underlining the importance of estrogen-mediated intestinal microbiota changes [11,13]. Additionally, a study on rats showed that several intestinal microbiota-derived microRNAs modulated steroid biosynthesis and estrogen signaling [12]. However, to our knowledge, reproductive hormone-mediated intestinal microbiota changes have not yet been explored in ruminants. Furthermore, functional changes in the intestinal microbiota mediated by reproductive hormones need to be clarified in ruminants.

Reproductive hormones change during estrus synchronization, and estrus is induced by estrogen [14,15]. Based on these previous studies, we hypothesized that changes in the intestinal microbiota occur during estrus synchronization and that these shifts in the intestinal microbiota are caused by reproductive hormones. Therefore, the objective of this study was to determine the shifts in the intestinal microbiota during estrus synchronization in grazing Simmental cows and to clarify the role of reproductive hormones in mediating these shifts in the intestinal microbiota.

## 2. Materials and Methods

All the study procedures were reviewed and approved by the Institutional Animal Care and Use Committee at Inner Mongolia Agricultural University, Hohhot, China (protocol no. 2020079). and were performed in accordance with the guiding principles of the Humane Treatment of Laboratory Animals (HTLA Pub. Chapter 2–6, revised 2006 in China).

### 2.1. Animals and Experimental Procedure

The study was conducted at the Prairie Chenbarhu Banner, Hulunbuir, Inner Mongolia, China (latitude 49.3° and longitude 119.4°). The study lasted for 12 days, beginning on 15 June 2021 and ending on 27 June 2021. The location has a mid-temperate semi-humid and semiarid continental climate, average relative humidity (25.04 ± 24.07%), and average temperature (18.96 ± 7.95 °C). The relative humidity and ambient temperature were measured daily in the morning (06:00 to 08:00), afternoon (13:00 to 15:00), and evening (18:00 to 20:00) and averaged from three different locations within the grassland. These data were measured every 10 min by Hobo Pro Series Temp probes (Onset Computer Corporation, Pocasset, MA, USA) during the entire experiment, and the probes were hung on the fence approximately 1.5 m above the ground. The cattle were selected from among healthy multiparous nonlactating Simmental cows (3 parities and weighing 550 ± 35 kg), and the body condition score was 5–7 (scale of 1–9, where 1 = very thin and 9 = very fat). The cattle were adapted to grazed grassland for two weeks before the experiment and allowed to graze naturally and drink water freely during the experiment; no shade or house was provided for the cows. The water was well water, and the grazed grassland mainly included *Stipa grandis*, *Leymus chinensis*, *Stipa baicalensis*, *Achnatherum sibiricum*, *Bupleurum scorzonerifolium*, and *Cleistogenes squarrosa*. The mixed herbage samples were collected and analyzed to determine their chemical compositions, and the methods and results of the herbage analyses are provided in Appendix A. All the cattle were in the normal estrous cycle, and the initial ovarian stage was that all the cattle had 1 or 2 dominant follicles (absence of corpus luteum) according to a rectal examination prior to initiation of the experiment, and all the cattle had healthy productive tracts (no metritis, cystitis, postpartum diseases, etc.). An estrus synchronization protocol of fixed-time artificial insemination was carried out in all the cattle. On day 0 (D0), all of the animals were treated with one intravaginally controlled internal drug release (CIDR) device (Cue-Mate^®^, Vetoquinol Australia Pty Ltd., Brisbane, QLD, Australia) containing 1.9 g of progesterone for nine days plus 5 mL of vitamin ADE, which was injected intramuscularly (Xixiang Changjiang Animal Medicine Co., Ltd., Hanzhong, China). Then, three cloprostenol intramuscular injections (0.2 mg/injection, Shanghai Jisheng Biotechnology Co., Ltd., Shanghai, China) were administered on D9 when the CIDR was withdrawn, and the experiment ended on D12.

### 2.2. Sample Collection and Analysis of Serum Reproductive Hormones

On D0, D9, and D12, at 10:00, blood samples were collected via coccygeal venipuncture into tubes containing no additives (Shijiazhuang Kang Wei Shi Medical Equipment Co., Ltd., Shijiazhuang, China). All the blood samples were harvested before each treatment on the sampling days; that is, blood samples were harvested before the CIDR was placed into the vagina of the cattle on D0, before the injected cloprostenol intramuscular injections on D9, and before artificial insemination on D12. The tubes were immediately placed on ice, cooled, and centrifuged at 1800× *g* for 10 min, and the serum was collected and transferred to the laboratory and stored at −80 °C. The serum was analyzed to measure the levels of gonadotropin-releasing hormone (GnRH), prolactin (PRL), oxytocin (OT) [16,17], follicle-stimulating hormone (FSH), luteinizing hormone (LH), progesterone (PROG), and estrogen 2 (estradiol, E2) [18] using bovine enzyme-linked immunosorbent assay kits (Shanghai Baomanbio Biotech Co., Ltd., Shanghai, China) according to the manufacturer’s instructions.

### 2.3. Sample Collection and Total DNA Extraction of Rectal Microbes

Fecal samples from the rectum were collected at the same time as blood samples, transferred into sterile and pyrogen-free centrifuge tubes, immediately frozen in liquid nitrogen, and stored at −80 °C for further genomic DNA extraction [19]. Total DNA extraction was performed using the hexadecyl trimethyl ammonium bromide (CTAB) method according to a previously described protocol [20]. A NanoDrop ND-1000 spectrophotometer (Thermo Fisher Scientific, Waltham, MA, USA) and agarose gel electrophoresis (1%) were used to assess the quantity and quality of the extracted DNA, respectively.

### 2.4. 16S rRNA Gene Sequencing

The V3–V4 hypervariable regions of the microbial 16S rRNA genes were amplified and sequenced with the Illumina MiSeq platform (Majorbio BioPham Technology, Shanghai, China) using the primers 338F (5′-ACTCCTACGGGAGGCAGCA-3′) and 806R (5′-GGACTACHVGGGTWTCTAAT-3′). PCR was performed in triplicate under the following conditions: initial denaturation at 95 °C for 3 min, 29 cycles of 95 °C for 30 s, 55 °C for 30 s, and 72 °C for 45 s, followed by a final extension at 72 °C for 10 min [20]. The PCR mixture included 5 × PrimeSTAR buffer (4 μL), dNTP (2.5 mM) 2 μL, forward primer (5 μM) 0.8 μL, reverse primer (5 μM) 0.8 μL, PrimeSTAR heat stress DNA polymerase 0.4 μL, and template DNA 20 ng. Two percent agarose gels were used to detect the PCR products, which were then purified using a DNA purification kit (Axygen, Biosciences, Union City, CA, USA). The raw 16S rRNA gene sequences were demultiplexed, quality-filtered with fastp (version 0.20.0, HaploX Biotechnology Co., Ltd., Shenzhen, China), and merged with FLASH (version 1.2.7, http://ccbjhu.edu/saftware/FLASH/ (accessed on 15 March 2022)); the sequences were quality-filtered using the reported criteria [19]. Operational taxonomic units (OTUs) were clustered based on a 97% similarity threshold using Uparse (version 7.0, http://drive5.com/uparse/ (accessed on 15 March 2022)) [21]. The taxonomy of each OTU was assigned by classifying its representative sequence using the RDP Classifier algorithm (http://rdp.cme.msu.edu/ (accessed on 12 November 2021)) against the SILVA 16S rRNA database (SSU123; https://www.arb-silva.de/ (accessed on 12 November 2021)) using a confidence threshold of 70% [22].

### 2.5. Microbial Data Processing

Alpha diversity was evaluated with the observed richness (Sobs), Chao1 richness estimator, ACE estimator, Shannon diversity index, and Simpson diversity index. Beta diversity was assessed using Bray–Curtis distances and analysis of similarities (ANOSIM) and visualized using the principal coordinate analysis (PCoA). ANOSIM is a nonparametric method used to test the differences in community structures among populations. We used linear discriminant analysis (LDA) with a LDA effect size (LEfSe) to identify taxa characterizing the differences among three metadata classes (phylogenetic levels from genus to phylum), and the default setting LDA score filter value was 3. A functional prediction analysis for the intestinal microbiota was carried out using the phylogenetic investigation of communities by reconstruction of unobserved states (PICRUSt) analysis, and the predicted functional composition profiles were collapsed into levels 1–3 of the Kyoto Encyclopedia of Genes and Genomes (KEGG) database pathways [23]. Only pathways that were significantly enriched (*p* < 0.05) are shown. The correlation coefficients were determined between the data generated from reproductive hormones of the serum and intestinal biomarker microbiota (the results of the LEfSe analysis) by Spearman’s correlation.

### 2.6. Statistical Analysis

The serum data of reproductive hormones were analyzed by one-way analysis of variance (ANOVA) followed by Duncan’s multiple comparison test (GLIMMIX procedure, SAS 9.2, SAS Institute Inc., Cary, NC, USA). Alpha diversity data were compared using a *t*-test, and the predicted KEGG pathways were compared using the Wilcoxon rank–sum test. Statistical analysis was performed using SAS. R packages (version 3.3.1) were used in the beta diversity and Spearman correlation analyses. Differences in the compared data were defined as significant at *p* < 0.05.

## 3. Results

### 3.1. Reproductive Hormone Shifts in Host Serum

The results for the reproductive hormones in cow serum are shown in Figure 1. There was no change in the GnRH level among the three sampling times (*p* > 0.05). The FSH, LH, and PRL levels at D12 were higher than those at D0 and D9 (*p* < 0.05). The OT level at D12 was only higher than that at D0 (*p* < 0.05) but not higher than that at D9 (*p* > 0.05). The PROG level was higher at D12 than at D0 and D9 (*p* < 0.05). The level of E2 at D12 was higher than that at D0 (*p* < 0.05).

### 3.2. Diversity Shifts and Composition Biomarkers of Intestinal Microbiota

There was no change in the alpha diversity indices (Figure 2A,D–F), including community coverage (coverage) and community richness (Sobs), for D0 vs. D9 vs. D12 in the intestinal microbiota (*p* > 0.05). However, the two community diversity indices (Chao1 and ACE) at D12 were decreased compared with those at D0 (*p* < 0.05). The PCoA and Bray–Curtis distances were used to examine the beta diversity of the intestinal microbiota. The closer the two community points were, the more similar the species compositions of the two communities were. As shown in the PCoA plot based on Bray–Curtis distances in Figure 2G–J, the beta diversity of the microbiota shifted markedly: D0 vs. D9 vs. D12 (R = 0.23, *p* = 0.001), D0 vs. D9 (R = 0.22, *p* = 0.001), D0 vs. D12 (R = 0.14, *p* = 0.001), and D9 vs. D12 (R = 0.34, *p* = 0.001).

The LEfSe analysis revealed shifts in the intestinal microbiota communities among D0, D9, and D12 (Figure 3A,B). There was a greater abundance on D0 (*p* < 0.05 and LDA cutoff >3.0) of the families *Bacteroidaceae*, *Hungateiclostridiaceae*, and *F082* and the genera *Bacteroides* and *norank_f__F082*. There was a greater abundance on D9 (*p* < 0.05 and LDA cutoff >3.0) of the phyla *Patescibacteria* and *Actinobacteriota*; classes *Coriobacteriia* and *Saccharimonadia*; orders *Christensenellales*, *Saccharimonadales*, and *Coriobacteriales*; families *Christensenellaceae*, *Muribaculaceae*, *Saccharimonadaceae*, and *Atopobiaceae*; and genera *Prevotellaceae_UCG-004*, *Christensenellaceae_R-7_group*, *norank_f__Ruminococcaceae*, *Ruminococcus_torques_group*, *norank_f__Muribaculaceae*, *Candidatus_Saccharimonas*, *Saccharofermentans*, *Agathobacter*, and *Olsenella*. There was a greater abundance on D12 (*p* < 0.05 and LDA cutoff >3.0) of the phyla *Verrucomicrobiota* and *Cyanobacteria*; classes *Vampirivibrionia*, *Negativicutes*, and *Verrucomicrobiae*’ orders *Gastranaerophilales*, *Acidaminococcales*, *Verrucomicrobiales*, and *unclassified_c__Clostridia*; families *norank_o__Gastranaerophilales*, *Acidaminococcaceae*, *Akkermansiaceae*, *and unclassified_c__Clostridia*; and genera *Phascolarctobacterium*, *norank_f__norank_o__Gastranaerophilales*, *norank_f__Oscillospiraceae*, *Akkermansia*, *Alloprevotella*, *unclassified_c__Clostridia*, and *UCG-002*.

### 3.3. Function Shifts in the Intestinal Microbiota

Levels 1–3 of the KEGG pathways (relative abundance, %) were tested. There were no changes among D0, D9, and D12 at level 1, and data on the functional shifts are shown in Table 1 (level 2) and Appendix A (level 3). Compared with D0 and D9, on D12, at level 2, the enriched pathways involved genetic information and processing and metabolism (*p* < 0.05); pathways involving the excretory system and cellular processes and signaling were enriched on D12 compared with D9 (*p* < 0.05) but not compared with D0 (*p* > 0.05). Pathways involving carbohydrate metabolism and immune system diseases were less enriched on D12 than on D0 and D9 (*p* < 0.05).

### 3.4. Correlation of Reproductive Hormones and Intestinal Diversity Indices and Microbiota Biomarkers

The correlation of reproductive hormones in the serum and intestinal microbiota is presented in Table 2. The serum GnRH, LH, PROG, PRL, and OT levels did not correlate with any alpha diversity or beta diversity indices (Table 2; *p* > 0.05). The serum FSH levels correlated with PC2 of the PCoA for beta diversity (R = −0.34, *p* = 0.026), and the E2 levels correlated with Chao1 for the alpha diversity (R = −0.38, *p* = 0.013) and PC1 of the PCoA for the beta diversity (R = −0.32, *p* = 0.048).

We did not present the data showing that the intestinal microbiota did not correlate with any of the seven reproductive hormones of the serum in Table 3 (uncorrelated results are presented in Appendix A). The serum GnRH levels did not correlate with any of the biomarkers of the intestinal microbiota (*p* > 0.05) (Appendix A). The serum FSH levels correlated (*p* < 0.05) with biomarkers at the phylum level (*Actinobacteriota*, *Verrucomicrobiota*, and *Patescibacteria*); class level (*Vampirivibrionia*, *Coriobacteriia*, *Verrucomicrobiae*, and *Saccharimonadia*); order level (*Saccharimonadales*, *Verrucomicrobiales*, and *Coriobacteriales*); family level (*Hungateiclostridiaceae*, *Akkermansiaceae*, *Atopobiaceae*, and *Saccharimonadaceae*); and genus level (*Ruminococcus_torques_group*, *Candidatus_Saccharimonas*, *Akkermansia*, *Saccharofermentans*, and *Olsenella*). The serum LH levels correlated (*p* < 0.05) with biomarkers at the phylum level (*Verrucomicrobiota*); class level (*Vampirivibrionia*, *Coriobacteriia*, and *Verrucomicrobiae*); order level (*Verrucomicrobiales* and *Coriobacteriales*); family level (*Akkermansiaceae* and *Atopobiaceae*); and genus level (*Ruminococcus_torques_group*, *Akkermansia*, *Saccharofermentans*, and *Olsenella*). The serum PROG levels correlated (*p* < 0.05) with biomarkers at the phylum level (*Verrucomicrobiota*), family level (*Atopobiaceae*), and genus level (*Olsenella*). The serum E2 levels correlated (*p* < 0.05) with biomarkers at the phylum level (*Actinobacteriota* and *Verrucomicrobiota*); class level (*Vampirivibrionia*, *Negativicutes*, *Verrucomicrobiae*, and *Saccharimonadia*); order level (*Acidaminococcales*, *Gastranaerophilales*, and *Verrucomicrobiales*); family level (*Acidaminococcaceae*, *Hungateiclostridiaceae*, *norank_o__Gastranaerophilales*, *Akkermansiaceae*, and *Atopobiaceae*); and genus level (*Phascolarctobacterium*, *norank_f__Ruminococcaceae*, *norank_f__norank_o__Gastranaerophilales*, *Akkermansia*, *Saccharofermentans*, and *Olsenella*); and the serum E2 levels correlated with *g__Ruminococcus_torques_group* (R = −0.51, *p* = 0.00049). The serum PRL levels correlated (*p* < 0.05) with biomarkers at the class level (*Vampirivibrionia* and *Verrucomicrobiae*), order level (*Verrucomicrobiales*), family level (*Hungateiclostridiaceae* and *Akkermansiaceae*), and genus level (*Akkermansia* and *Saccharofermentans*). The serum OT levels correlated (*p* < 0.05) with biomarkers at the order level (*Christensenellales*), family level (*Christensenellaceae*), and genus level (*Christensenellaceae_R-7_group*).

## 4. Discussion

In recent years, characterization of the microbiota changes in livestock animals has been of great interest to improve the reproductive performance. Even though synchronization protocols have been widely used and the microbiota of the vagina and uterus have been studied, the intestinal microbiota shifts in hosts undergoing these protocols are largely unknown. To better comprehend the synchronization protocol in cattle, we explored the intestinal microbiota of ruminants by using a grazing Simmental cow model. Our hypothesis was that intestinal microbiota shifts associated with the synchronizing protocol are caused by changes in the reproductive hormones.

It is known that the diversity and succession of microbes change with the host reproductive tract health status and reproductive status in cattle [24,25]. In the present study, the diversity and function clearly showed a shift in the intestinal microbiota during synchronization, and decreased alpha diversity indices (Chao1 and ACE; D12 vs. D0) were detected. This result was consistent with the decreased alpha diversity of the uterine and vaginal microbiota of estrus cattle in the synchronizing protocol [25]. Additionally, the intestinal microbiota changed significantly during conception [26]. All these studies showed the intestinal microbiota adjustments in response to the host reproductive status, and the decreased intestinal microbiota diversity may allow the host to more potentially respond to changes through the estrous cycle to prepare for host estrous [25,26]. On the other hand, the reproductive microbiota has a low diversity, with an increase in diversity leading to disease and fertility issues in humans [27,28]. We observed a decrease in the immune system diseases pathway, and we also observed a decrease in the carbohydrate digestion and absorption pathway, a level 3 KEGG pathway (Appendix A). These results may also provide evidence for the decreased alpha diversity indices in the present study.

Host estrus is characterized by hormone fluctuations especially increasing the E2 concentration in blood [14,15]. This was supported by the results in the present study, whereby the host estrus state was associated with an increased E2 concentration in the serum when compared with the non-estrus state (D12 vs. D0). These results were supported by our result that the intestinal microbiota on D12 was associated with enriched steroid hormone biosynthesis, steroid biosynthesis, and arachidonic acid metabolism pathways (Appendix A). Steroid hormone biosynthesis and steroid biosynthesis contribute to the biosynthesis of reproductive hormones [29,30], and arachidonic acid is used as a biosynthetic precursor for prostaglandins [31].

The function of estrogen in influencing the intestinal microbiota structure in humans has been shown [9,10], and the underlying mechanism is the direct modulation of the metabolism of the microbiota through estrogen receptor beta [11,13]. Simultaneously, the intestinal microbiota with beta-glucuronidase activity deconjugates the conjugated circulating estrogen excreted in the bile, and beta-glucuronidase reduces the inactivation of estrogen and increases the estrogen concentration in the circulation of the host body, so the intestinal microbiota is one principal regulator of circulating estrogen [10,11,32]. The present study provided strong evidence that E2 (E2 is one main estrogen hormone) correlated with the abundance of the biomarkers and alpha and beta diversity of the intestinal microbiota. In humans, the levels of total urinary estrogen are very strongly and directly associated with intestinal microbiota alpha diversity (Sobs and Shannon indices) [33]. Our results showed that E2 is related to *Ruminococcaceae* (*g__norank_f__Ruminococcaceae*). This correlation between *Ruminococcaceae*, which produces beta-glucuronidase, and estrogen was reported previously [33]. In addition, *Ruminococcaceae* is a family of autochthonous and benign species [34] that produce short-chain fatty acids and has been well-illustrated to be responsible for the degradation of diverse polysaccharides and fibers [35]. The population size of *Ruminococcaceae* is inversely correlated with hepatic encephalopathy, nonalcoholic fatty liver disease, and systemic lupus erythematosus [35,36,37], and we detected a decrease in the immune system diseases pathway, which may be due to this bacterium. In the present study, we reported that E2 correlated with the *g__Ruminococcus_torques_group*. The *g__Ruminococcus_torques_group*, which belongs to the family *Lachnospiraceae*, was found to be related to fat deposition [38,39]. A positive association of abdominal fat and subcutaneous fat thickness with *f__Lachnospiraceae* abundance was observed in the broilers [40]. Our results showed that host estrus resulted in an increase in intestinal microbiota function in the biosynthesis of unsaturated fatty acids and fatty acid elongation in the mitochondria (Appendix A).

The reproductive hormones FSH, LH, PROG, PRL, and OT correlated with some biomarkers of the intestinal microbiota of the host in the present study. Additionally, the relationship between these reproductive hormones and the intestinal microbiota is present in other animals and human models [10,11,41], possibly owing to the intestinal microbiota mediated in the process of metabolizing these hormones [10,11]. However, we could not determine whether GnRH was correlated with any of the biomarkers and diversity indices of the intestinal microbiota in the present study. The reason may be that this hormone is produced in the hypothalamus only or that there is no receptor for this hormone among the intestinal microbiota [11]. In addition, the high levels of PROG at D12 did not decrease compared with those at D9 in the present study. We observed the estrus of cows in the afternoon of D11. Generally, cows ovulate 10–12 h after estrus, and the corpus luteum forms and releases PROG to the body at the same time as ovulation. Therefore, serum was collected at D12 before artificial insemination; at that time, the cows ovulating, which led to an increasing trend of the PROG levels.

The present study provided references for ruminant intestinal shifts during estrus synchronization; however, other physiological statuses of intestinal microbiota shifts during estrus synchronization in other ruminants, such as lactating dairy cows that exhibit milking-induced prolactin release [42], are of interest and should be determined. In addition, other protocols of estrus synchronization, not a protocol of vaginal progesterone-controlled release plus cloprostenol in the present study, may also make the ruminant intestinal microbiota shift and should be investigated in future studies. In the present study, the functions of metabolism and immunity of the intestine microbiota shifted during estrus synchronization, which may indicate that the same shifts occurred for the host, given the well-known interaction between the host and intestine microbiota [43]. This inference is supported by the fact that the animals who were in estrus also exhibited an increase in metabolism and immune functions [44,45,46]. Furthermore, biomarkers of the intestine microbiota, such as *Ruminococcaceae*, which correlated with the reproductive hormone estradiol during estrus synchronization, should be studied further and may be useful biomarkers to detect cattle estrus.

## 5. Conclusions

The structure (alpha and beta diversity indices), composition, and function of the intestinal microbiota shifted during estrus synchronization in grazing Simmental cows, and these shifts were mediated by reproductive hormones—in particular, by estradiol. The functions of metabolism and immunity were enriched in the intestinal microbiota during estrus synchronization. All these results provide information about estrus synchronization in cattle, and some nutritional and immune-enhancing strategies may be applied during synchronization based on the present results.

## Figures and Tables

**Figure 1 animals-12-01751-f001:**
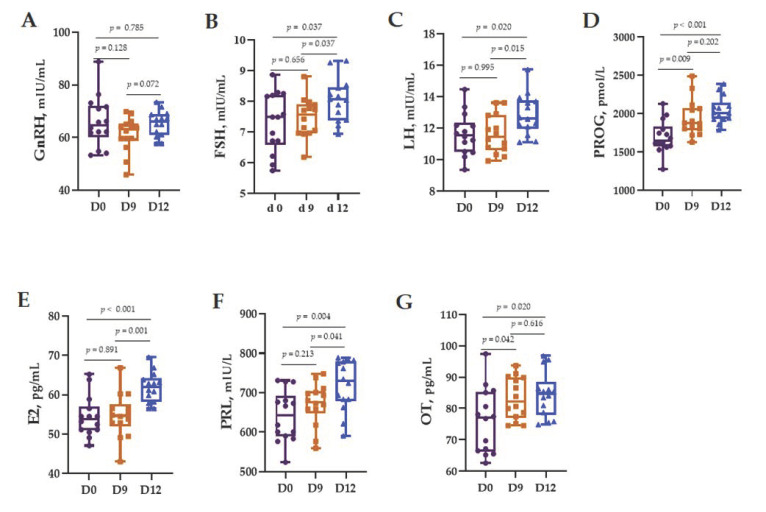
Serum changes in the reproductive hormones on day 0 (D0), day 9 (D9), and day 12 (D12). GnRH: gonadotropin-releasing hormone (**A**), FSH: follicle-stimulating hormone (**B**), LH: luteinizing hormone (**C**), PROG: progesterone (**D**), E2: estrogen 2 (**E**), PRL: prolactin (**F**), and OT: oxytocin (**G**). Differences were defined as significant at *p* < 0.05.

**Figure 2 animals-12-01751-f002:**
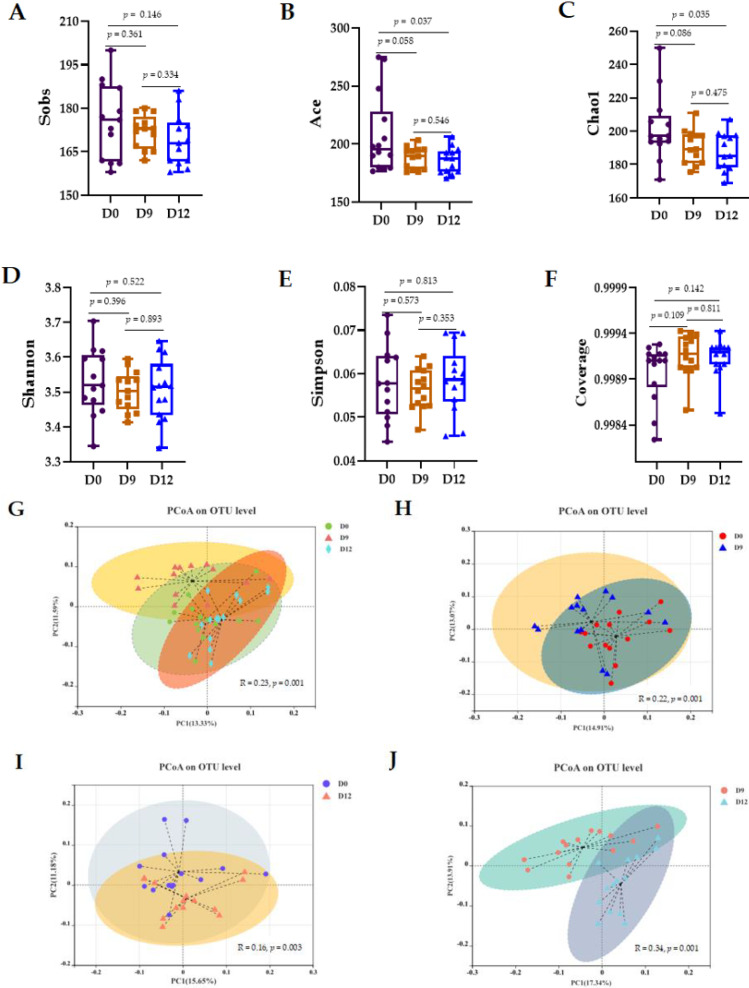
Shifts in the alpha diversity and beta diversity of the intestinal microbiota. Alpha diversity was evaluated with the observed richness (Sobs, (**A**)), ACE estimator (**B**), Chao1 richness estimator (**C**), Shannon diversity index (**D**), Simpson diversity index (**E**), and coverage (**F**). Beta diversity is shown with a principal coordinate analysis (PCoA) plot based on community membership as measured by the Bray–Curtis distances. D0 vs. D9 vs. D12 (**G**), D0 vs. D9 (**F**), D0 vs. D12 (**I**), and D9 vs. D12 (**J**). PC1 = principal component 1; PC2 = principal component 2. The ellipses were calculated and drawn with a 0.95 confidence level. R value represents the similarity between groups, and the greater the value, the lower the similarity of sample groups. Differences were defined as significant at *p* < 0.05.

**Figure 3 animals-12-01751-f003:**
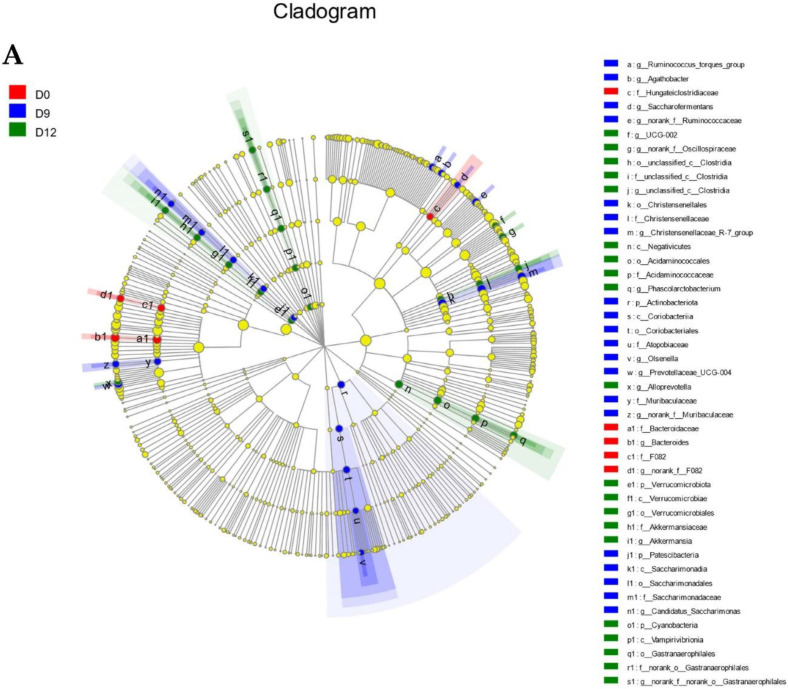
Biomarker analysis of each taxon of the intestinal microbiota. (**A**) Cladogram of the linear discriminant analysis (LDA) effect size (LEFSe) analysis shows the overrepresented microbial populations. Circles from the inside out indicate the phylogenetic levels from the phylum to genus. (**B**) Taxa are significant from LEFSe (*p* < 0.05 and the LDA cutoff >3.0). Colored nodes from the center to the periphery represent the phylum (p), class (c), order (o), family (f), and genus (g) level differences detected among D0 (red), D9 (blue), and D12 (green).

**Table 1 animals-12-01751-t001:** The significantly enriched predicted level 2 KEGG pathways (relative abundance, %) on day 0 (D0), day 9 (D9), and day 12 (D12).

Pathway Description	D0	D9	D12	SEM	*p*-Value
Carbohydrate metabolism	10.105 ^a^	10.137 ^a^	10.067 ^b^	0.0117	<0.001
Excretory system	0.0234 ^ab^	0.0231 ^b^	0.0248 ^a^	0.00048	0.038
Cellular processes and signaling	3.984 ^a^	3.957 ^b^	3.989 ^a^	0.0063	0.002
Genetic information processing	2.788 ^b^	2.779 ^b^	2.808 ^a^	0.0047	<0.001
Immune system diseases	0.0295 ^a^	0.0296 ^a^	0.0281 ^b^	0.00039	0.046
Metabolism	2.426 ^b^	2.423 ^b^	2.441 ^a^	0.00340	0.001

SEM = standard error of the means. ^a,b^ Means without a common superscript within a row differ significantly (*p* < 0.05).

**Table 2 animals-12-01751-t002:** The relationship of reproductive hormones and diversity indices of the intestinal microbiota.

Item	GnRH	FSH	LH	PROG	E2	PRL	OT
Sobs	−0.30	−0.28	0.01	−0.17	−0.24	−0.26	0.60
Shannon	0.15	−0.19	0.06	−0.05	−0.06	−0.27	−0.01
Simpson	−0.03	0.22	−0.04	−0.05	0.07	0.22	−0.04
Ace	−0.09	−0.31	0.06	−0.19	−0.29	−0.07	0.04
Chao1	−0.08	−0.30	0.08	−0.11	−0.32 *	−0.11	−0.08
Coverage	0.02	−0.11	−0.09	−0.01	−0.15	0.27	−0.19
PC1	0.29	0.28	0.26	0.28	0.38 *	0.02	0.11
PC2	−0.12	−0.34 *	−0.04	0.17	−0.27	−0.12	0.24

* *p* < 0.05. Alpha diversity indices: the observed richness (Sobs), Chao1 richness estimator, ACE estimator, Shannon diversity index, and Simpson diversity index. Beta diversity indices: PC1: principal component 1; PC2: principal component 2. GnRH: gonadotropin-releasing hormone, FSH: follicle-stimulating hormone, LH: luteinizing hormone, PROG: progesterone, E2: estrogen 2, PRL: prolactin, and OT: oxytocin. The correlation uses Spearman’s rank correlation coefficient and 2–tailed significance test (*n* = 42).

**Table 3 animals-12-01751-t003:** The relationship of the reproductive hormones and biomarker levels of the intestinal microbiota.

Item	FSH	LH	PROG	E2	PRL	OT
*p__Actinobacteriota*	−0.36 *	−0.24	−0.13	−0.33 *	−0.10	0.05
*p__Verrucomicrobiota*	0.36 *	0.36 *	0.32 *	0.44 **	0.30	0.26
*p__Patescibacteria*	−0.41 **	−0.20	−0.03	−0.30	−0.22	0.02
*o__Acidaminococcales*	0.27	0.22	0.08	0.38 *	0.06	0.01
*o__Christensenellales*	0.03	0.11	0.15	0.16	0.08	0.40 **
*o__Saccharimonadales*	−0.41 **	−0.20	−0.03	−0.30	−0.22	0.02
*o__Gastranaerophilales*	0.29	0.29	0.20	0.31 *	0.13	0.12
*o__Verrucomicrobiales*	0.34 *	0.31 *	0.28	0.41 **	0.31 *	0.27
*o__Coriobacteriales*	−0.32 *	−0.31 *	−0.20	−0.27	−0.12	0.00
*c__Vampirivibrionia*	0.34 *	0.31 *	0.28	0.41 **	0.31 *	0.27
*c__Negativicutes*	0.25	0.21	0.07	0.38*	0.06	0.01
*c__Coriobacteriia*	−0.32 *	−0.31 *	−0.20	−0.27	−0.12	0.00
*c__Verrucomicrobiae*	0.34 *	0.31 *	0.28	0.41 **	0.31 *	0.27
*c__Saccharimonadia*	−0.41 **	−0.20	−0.03	−0.30	−0.22	0.02
*f__Acidaminococcaceae*	0.27	0.22	0.08	0.38 *	0.06	0.01
*f__Christensenellaceae*	0.03	0.11	0.15	0.16	0.08	0.40 **
*f__Hungateiclostridiaceae*	−0.43 **	−0.21	−0.21	−0.35 *	−0.37 *	0.07
*f__norank_o__Gastranaerophilales*	0.29	0.29	0.20	0.31 *	0.13	0.12
*f__Akkermansiaceae*	0.34 *	0.31 *	0.28	0.41 *	0.31 *	0.27
*f__Atopobiaceae*	−0.35 *	−0.35 *	−0.36 *	−0.38 *	−0.16	−0.07
*f__Saccharimonadaceae*	−0.41 **	−0.20	−0.03	−0.30	−0.22	0.02
*g__Ruminococcus_torques_group*	−0.42 **	−0.32 *	−0.29	−0.51 ^#^	−0.17	−0.22
*g__Phascolarctobacterium*	0.27	0.23	0.09	0.39 *	0.07	0.01
*g__Christensenellaceae_R-7_group*	0.02	0.10	0.16	0.15	0.11	0.40 **
*g__norank_f__Ruminococcaceae*	−0.16	−0.03	−0.05	−0.31 *	0.10	0.13
*g__Candidatus_Saccharimonas*	−0.41 **	−0.20	−0.03	−0.30	−0.22	0.02
*g__norank_f__norank_o__Gastranaerophilales*	0.29	0.29	0.20	0.31 *	0.13	0.12
*g__Akkermansia*	0.34 *	0.31 *	0.28	0.41 **	0.31 *	0.27
*g__Saccharofermentans*	−0.37 *	−0.30	−0.30	−0.44 **	−0.40 **	0.06
*g__Olsenella*	−0.34 *	−0.35 *	−0.37 *	−0.40 **	−0.20	−0.10

* *p* < 0.05, ** *p* < 0.01, and ^#^
*p* < 0.001. GnRH: gonadotropin-releasing hormone, FSH: follicle-stimulating hormone, LH: luteinizing hormone, PROG: progesterone, E2: estrogen 2, PRL: prolactin, and OT: oxytocin. The correlation using Spearman’s rank correlation coefficient and the 2–tailed significance test (*n* = 42).

## Data Availability

The datasets that were applied and/or analyzed throughout the prevailing research are available from the corresponding author.

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
