# Peer review of "Reproductive Hormones Mediate Intestinal Microbiota Shifts during Estrus Synchronization in Grazing Simmental Cows"

_animals, 2022, doi:10.3390/ani12141751_

Round 1
Reviewer 1 Report
Minor comments:
- ln 86-88. how did you measure humidity and temperature. Please specify.
- ln 92 describe deeply feeding. Any feed or water analyses? Maybe they could influence on micriome variations
- ln 93. on day 0 did you perform a p4 analyses? what was the inital ovarian and reproductive tract stage?
- ln 96. 5ml is not appropriate; please mg/kg or something like that
- ln 98. pgf dose?
- ln 100. reference of this protocol. It is an old one, why did you use this?
- ln 109. did you measure all by ELISA? Please provide references which support the usefulness of ELISA with every hormone.
- ln 134. please here and in every part of the manuscript, take care of format references. Maybe just the number and in the reference list place the link or url. The same in ln 135, 136, and so on.
- avoid starting any sentence with acronyms.
- figures are copy-pasted from word or another software, please follow journal guidelines. I can not visualize them properly. Apply this comment to every figure and graphic
- figure 1. how could you explain higher levels of p4 at the end of the protocol?
- table 3, maybe include just what has a significant correlation?
- discussion: any difference due to breed? reproductive protocol? i woudl try to include a paragraph focusing on importance in the daily veterinary routine of these findings.
- missing the number of ethic committee aproval.
- references: double numbered in the reference list. Please check
- in general, take care of the format, please
Author Response
Overall response from the authors:
We thank you for your work; all the comments and suggestions were constructive. The entire manuscript was considerably improved when we made the corresponding revisions. These revisions were made mainly in the Materials and methods section and at the end of the Discussion section. We also provided point-by-point response to all the minor comments. We thank you again for your work on our manuscript, which enabled us to think about and improve our research. Thank you very much, and please reconsider our manuscript.
Minor comments:
Point 1: ln 86-88. how did you measure humidity and temperature. Please specify.
Response 1: Thank you for your suggestion. We explained how we measured humidity and temperature in Lines 92–98 in the revised manuscript as follows:
The relative humidity and ambient temperature were measured daily in the morning (06:00 to 08:00), afternoon (13:00 to 15:00), and evening (18:00 to 20:00) and averaged from three different locations within grassland. These data were measured every 10 min by Hobo Pro Series Temp probes (Onset Computer Corporation, Pocasset, Massachusetts, USA) during the entire experiment, and the probes were hung on the fence approximately 1.5 m above the ground.
Point 2: ln 92 describe deeply feeding. Any feed or water analyses? Maybe they could influence on micriome variations
Response 2: We performed an herbage analysis but not a water analysis, and the methods and results of the herbage analysis are shown in the supplementary materials and Table S1.
Since the cows were grazing in the same grassland, the same well water and herbage were provided to the cows, and no supplementary feed was provided during the experiment, we could exclude the effect of diet on variations in the microbiome.
In addition, we thoroughly described how the grazing cows fed in Lines 100–107 in the revised manuscript as follows:
The cattle were adapted to grazed grassland for two weeks before the experiment and allowed to graze naturally and drink water freely during the experiment; no shade or house was provided for the cows. The water was well water, and the grazed grassland mainly included Stipa grandis, Leymus chinensis, Stipa baicalensis, Achnatherum sibiricum, Bupleurum scorzonerifolium, and Cleistogenes squarrosa. The mixed herbage samples were collected and analyzed to determine their chemical compositions, and the methods and results of the herbage analyses are provided in the supplementary materials and Table S1.
Point 3: ln 93. on day 0 did you perform a p4 analyses? what was the initial ovarian and reproductive tract stage?
Response 3: We performed P4 analyses on day 0.
All the cattle were in the normal estrous cycle, and the initial ovarian stage was that all the cattle had 1 or 2 dominant follicles (absence of corpus luteum) according to a rectal examination prior to initiation of the experiment, and all the cattle had healthy productive tracts (no metritis, kysthitis, postpartum diseases, etc.).
We have also added this information to Lines 107–111 in the revised manuscript.
Point 4: ln 96. 5ml is not appropriate; please mg/kg or something like that
Response 4: We ensured that 5 mL of vitamin ADE was injected into each cow, and we revised the sentence in Line 115 for clarity as follows: plus 5 mL of vitamin ADE, which was injected intramuscularly.
Point 5: ln 98. pgf dose?
Response 5: Yes, cloprostenol is an analog of prostaglandin F2 α, and the dose of cloprostenol was adopted from that of prostaglandin F2 α.
Point 6: ln 100. reference of this protocol. It is an old one, why did you use this?
Response 6: We selected this protocol because it is a classic protocol and has a higher success rate for inducing estrus with a vaginal progesterone release protocol. Of course, it is of interest for us to determine whether the intestinal microbiota shifts after the implementation of other protocols, such as GnRH injection and FSH injection.
Point 7: ln 109. did you measure all by ELISA? Please provide references which support the usefulness of ELISA with every hormone.
Response 7: Yes, we measured the levels of all hormones by ELISA, and we provide references here (Swelum et al., 2015; Salci et al., 2018; Moronkeji et al., 2021) and in the revised manuscript (Lines 128–133) as follows:
The serum was analyzed to measure the levels of gonadotropin-releasing hormone (GnRH), prolactin (PRL), oxytocin (OT) [16.17], follicle-stimulating hormone (FSH), luteinizing hormone (LH), progesterone (PROG), and estrogen 2 (estradiol, E2) [18] using bovine enzyme-linked immunosorbent assay kits (Shanghai Baomanbio Biotech Co., Ltd, Shanghai, China) according to the manufacturer’s instructions.
References:
Salci ESO, Demirbilek SK, Gunes N, Goncagul G, Uzabacı E, Carli T, Seyrek-Intas K. Comparison of the endocrinological and immunological results of different induction of parturition methods in ewes. Tierarztl Prax Ausg G Grosstiere Nutztiere. 2018;46(1):22-28. doi: 10.15653/TPG-170136.
Moronkeji MA, Emokpae MA, Ojo TA, Moronkeji RE, Ogundoju LT. The patterns and occupational distribution of hormonal abnormalities among men investigated for infertility in some centers in the southwest, Nigeria. J Clin Transl Res. 2021; 12;7(2):221-228.
Swelum AA, Alowaimer AN, Abouheif MA. Use of fluorogestone acetate sponges or controlled internal drug release for estrus synchronization in ewes: Effects of hormonal profiles and reproductive performance. Theriogenology. 2015; 1;84(4):498-503. doi: 10.1016/j.theriogenology.2015.03.018.
Point 8: ln 134. please here and in every part of the manuscript, take care of format references. Maybe just the number and in the reference list place the link or url. The same in ln 135, 136, and so on.
Response 8: Thank you for your suggestion. We reviewed the formatting of the references in the journal Animals, and all the links or URLs were added to the manuscript are applicable. We can see the published reference as shown below:
Li H, Ma L, Li Z, Yin J, Tan B, Chen J, Jiang Q, Ma X. Evolution of the Gut Microbiota and Its Fermentation Characteristics of Ningxiang Pigs at the Young Stage. Animals (Basel). 2021 Feb 27;11(3):638. doi: 10.3390/ani11030638.
Point 9: avoid starting any sentence with acronyms.
Response 9: All the sentences that started with acronyms were revised. These revisions can be found in Lines 263, 267, 271, 318, 332, 333, 345, 347, 353, 357, 367, 368, and 370.
Point 10: figures are copy-pasted from word or another software, please follow journal guidelines. I can not visualize them properly. Apply this comment to every figure and graphic
Response 10: All the figures were replaced with clear ones.
Point 11: figure 1. how could you explain higher levels of p4 at the end of the protocol?
Response 11: Thank you for your question. First, we re-evaluated our original data, and we confirmed that on D12, P4 remained at a level that was equal to that on D9. The data on D12 represent the results of the samples harvested before artificial insemination, and we provided an explanation, which we think will answer your questions, as follows:
When the CIDR was placed in the vagina on D0, a higher level of P4 was released in the cows, and P4 maintained the growth of the corpus luteum. Additionally, P4 could not maintain the growth of the corpus luteum since the amount of P4 decreased after several days, and the corpus luteum regressed to a normal level on D9. At the same time, GnRH, which is the hormones released from the hypothalamus, stimulated the pituitary gland to release FSH and LH so that E2 could be produced after follicular development and promote estrus in cows. We observed the estrus of cows in the afternoon of D11. Generally, cows ovulate 10–12 h after estrus, and the corpus luteum forms and releases P4 to the body at the same time as ovulation. Therefore, serum was collected at D12 before artificial insemination; at this time, cows had ovulated, which led to an increasing trend of P4 levels.
We did not measure P4 levels between D0 and D9, but the highest concentration of P4 in blood can reach several times the initial value, and this level can be achieved with other CIDR protocols (Bhoraniya et al., 2012; Kornmatitsuk et al., 2021). The concentration of P4 in the blood might decrease quickly after D9 when the CIDR is removed because of the short half-life (t½) (approximately 30 min) and high metabolic clearance rate (approximately 2.0%/min) of P4 (Hutchinson et al., 2012). Therefore, the concentration of P4 might be lower between D9 and D12 than that we determined on D9 and D12 and it might increase on D12 owing to the corpus luteum formation and release.
In addition, we added some information about sampling times to Lines 123–126 and discussion to Lines 447–452 of the revised manuscript.
References:
Bhoraniya HL, Dhami AJ, Naikoo M, Parmar BC, Sarvaiya NP. Effect of estrus synchronization protocols on plasma progesterone profile and fertility in postpartum anestrous Kankrej cows. Trop Anim Health Prod. 2012 Aug;44(6):1191-7. doi: 10.1007/s11250-011-0057-1.
Kornmatitsuk B, Kornmatitsuk S. Circulating progesterone concentrations and preovulatory follicle diameters affecting ovulatory response in crossbred dairy heifers, following a 7-day progesterone-based synchronization protocol. Trop Anim Health Prod. 2021 Jan 8;53(1):102. doi: 10.1007/s11250-020-02494-1.
Hutchinson IA, Dewhurst RJ, Evans AC, Lonergan P, Butler ST. Effect of grass dry matter intake and fat supplementation on progesterone metabolism in lactating dairy cows. Theriogenology. 2012 Sep 1;78(4):878-86. doi: 10.1016/j.theriogenology.2012.04.001.
Point 12: table 3, maybe include just what has a significant correlation?
Response 12: Thank you for your suggestion. We moved the data about the correlation of GnRH and the microbiota biomarker levels to Table S2 because of the nonsignificant correlation.
Point 13: discussion: any difference due to breed? reproductive protocol? i would try to include a paragraph focusing on importance in the daily veterinary routine of these findings.
Response 13: Thank you for your suggestion;, we find this suggestion constructive.
It is true that we did not discuss the limitations of the article related to the cattle’s breed and reproductive protocol, and we added these limitations to Lines 453–459 of the revised manuscript.
We added relative contents to the last paragraph with a focus on the importance of these findings for daily veterinary routines (Lines 459–468) in the revised manuscript.
Point 14: missing the number of ethic committee approval.
Response 14: The number of ethics committee approval (protocol No. 2020079) was added to the Materials and Methods section (Line 83) and Institutional Review Board Statement (Line 496).
Point 15: references: double numbered in the reference list. Please check
Response 15: Thank you for this reminder. We made these corrections in the reference list.
Point 16: in general, take care of the format, please
Response 16: Thank you for your suggestion. We carefully reviewed the Instructions for Authors of the journal and formatted our manuscript according to the requirements.

Reviewer 2 Report
The present study describes the intestinal microbiota and the modifications that occur during estrus synchronization in cows. The data obtained are original and can be useful to better understand the influence that estrus has on the microbiota and the implications in some pathologic processes. The manuscript is clearly written. Statistical analysis is well made too.
Major concerns:
1. Materials and methods. In cows that did not undergo synchronization how did the microbiota act? Remained the same or it underwent some modification? A comparison with a control group in the same breeding condition should be added to better understand modifications that occur.
Minor concerns:
1. Line 27. The first time an abbreviation is used the full name should be indicated. The full name of ACE and Chao1 are not indicated throughout the manuscript. The full name can be added into the table of abbreviation at the end of the manuscript.
2. Line 91. Body Condition Score is a scale with whole numbers. Decimal values should be removed to leave the whole number.
3. Figure 2. Graphical representations of 2B and 2C are switched in the caption. Please correct
4. Figure 3. Names of the bacteria are not visible in both figure 3A and 3B. If possible, make the legend more visible.
5. Table 2. In this table there are not values with a P value ≤ 0.05. The indication of P <0.01 and P<0.001 in this table is not needed and can be removed.
6. Line 269. Single sentences should be avoided in a scientific article. I recommend to change as follows: “and Olsenella); g__Ruminocossus_torques_group in particular showed a strong correlation with E2 (R=-0.51, P=0.00049).”
Author Response
Overall response from authors:
We thank you for your important suggestions about the experimental design and details in the manuscript, and we made corresponding revisions in response to each point. These changes have made our manuscript more readable. Thank you again for your contribution, and please reconsider our manuscript.
Major concerns:
1.Materials and methods. In cows that did not undergo synchronization how did the microbiota act? Remained the same or it underwent some modification? A comparison with a control group in the same breeding condition should be added to better understand modifications that occur.
Response: The microbiota always correlates with hormone levels regardless of whether the host is undergoing estrus synchronization. However, the microbiota will not shift when the host does not undergo synchronization. For example, there is a higher abundance of Ruminococcaceae when the host is in estrus since estrogen levels are high, and there is a lower abundance of the bacterium when estrogen levels are low (no estrus).
Thank you for your good comments and suggestion, and it is true that we did not include a blank control but only included a self-control (D0). However, for the purpose of assessing shifts in the intestinal microbiota, the relationship between hormone levels and intestinal microbiota has meaningful value.
It is generally believed that the intestinal microbiota is relatively stable in adulthood (the microbiota matures), its abilities to avoid disruption and self-regulate are very strong (Clemmons et al., 2019; Arshad et al., 2021), and the host genome can control the microbiome (Abbas et al., 2020). Therefore, it can be considered that the intestinal microbiota of cattle that do not undergo simultaneous synchronization will not change when they consume the same herbage and water. Therefore, please reconsider our work.
References:
Arshad MA, Hassan FU, Rehman MS, Huws SA, Cheng Y, Din AU. Gut microbiome colonization and development in neonatal ruminants: Strategies, prospects, and opportunities. Anim Nutr. 2021 Sep;7(3):883-895. doi: 10.1016/j.aninu.2021.03.004.
Clemmons BA, Voy BH, Myer PR. Altering the gut microbiome of cattle: considerations of host-microbiome interactions for persistent microbiome manipulation. Microb Ecol 2019;77:523e36. doi: 10.1007/s00248-018-1234-9.
Abbas W, Howard JT, Paz HA, Hales KE, Wells JE, Kuehn LA, Erickson GE, Spangler ML, Fernando SC. Influence of host genetics in shaping the rumen bacterial community in beef cattle. Sci Rep 2020;10:1e4. doi: 10.1038/s41598-020-72011-9.
Minor concerns:
1.Line 27. The first time an abbreviation is used the full name should be indicated. The full name of ACE and Chao1 are not indicated throughout the manuscript. The full name can be added into the table of abbreviation at the end of the manuscript.
Response 1: Thank you for your suggestion. The ACE and Chao1 indices were used to assess bacterial diversity, and no full names are available for these characteristic indices. In addition, we can see that the full names of the ACE and Chao1 indices were not indicated in the published references in journal Animals, as shown below:
Zhang Z, Huang B, Shi X, Wang T, Wang Y, Zhu M, Wang C. Comparative Analysis of Bacterial Diversity between the Liquid Phase and Adherent Fraction within the Donkey Caeco-Colic Ecosystem. Animals (Basel). 2022 Apr 26;12(9):1116. doi: 10.3390/ani12091116.
Li H, Ma L, Li Z, Yin J, Tan B, Chen J, Jiang Q, Ma X. Evolution of the Gut Microbiota and Its Fermentation Characteristics of Ningxiang Pigs at the Young Stage. Animals (Basel). 2021 Feb 27;11(3):638. doi: 10.3390/ani11030638.
In addition, we rechecked the abbreviation for other indices in the manuscript and the table of abbreviations.
2.Line 91. Body Condition Score is a scale with whole numbers. Decimal values should be removed to leave the whole number.
Response 2: Thank you for your suggestion. We made a correction (the average value was changed to the range of body condition scores, i.e., 6.05 ± 0.34 was changed to 5–7) according to your suggestion (Line 100).
3.Figure 2. Graphical representations of 2B and 2C are switched in the caption. Please correct
Response 3: Thank you very much, and we made this correction (Line 256).
4.Figure 3. Names of the bacteria are not visible in both figure 3A and 3B. If possible, make the legend more visible.
Response 4: Thank you for your suggestion. We made corrections to all the figures, which were replaced with clear figures (figures 1, 2, and 3).
5.Table 2. In this table there are not values with a P value ≤ 0.05. The indication of P <0.01 and P<0.001 in this table is not needed and can be removed.
Response 5: Thank you for your suggestion. We removed these p values (Table 2)
6.Line 269. Single sentences should be avoided in a scientific article. I recommend to change as follows: “and Olsenella); g__Ruminocossus_torques_group in particular showed a strong correlation with E2 (R=-0.51, P=0.00049).”
Response 6: Thank you for your suggestion. We made this correction according to your suggestion (Line 365).

Round 2
Reviewer 2 Report
Authors fulfilled all the comments made. The manuscript has a good readability and contents are presented in a good way. I think that the manuscript is suitable for publication on Animals.